

# Coupling effect of key factors on ecosystem services in border areas: a study of the Pu'er region, Southwestern China

Ruijing Qiao[1,2,3], Jie Li[1,2,3], Xiaofei Liu[1,2,3], Mengjie Li[4], Dongmei Lei[5], Yungang Li[1,2,3], Kai Wu[1,2,3], Pengbo Du[1,2,3], Kun Ye[1,2,3] and Jinming Hu[1,2,3]

[1] Institute of International Rivers and Eco-security, Yunnan University, Kunming, China
[2] Asian International Rivers Centre, Yunnan University, Kunming, China
[3] Yunnan Key Laboratory of International Rivers and Transboundary Eco-security, Yunnan University, Kunming, China
[4] School of Ecology and Environmental Science, Yunnan University, Kunming, China
[5] Yunnan University of Finance and Economics, Kunming, China

Corresponding authors
Jie Li, jie_li@ynu.edu.cn
Jinming Hu, hujm@ynu.edu.cn

## ABSTRACT

The coupling effects created by transboundary and local factors on ecosystem services are often difficult to determine. This poses great challenges for ecosystem protection and management in border areas. To decrease uncertainty, it is crucial to quantify and spatialize the impact multiple factors have on ecosystem services within different scenarios. In this study, we identified key transboundary and local factors from a set of 15 sorted factors related to four main ecosystem services. We employed a Bayesian Network—Geographic Information System (BN-GIS) model to simulate 90 scenarios with multiple factors combinations, quantifying and spatializing the coupling effects on the main ecosystem services. These simulations were conducted in the Pu'er region, which is situated alongside three countries, and serves as a representative border area in southwest China. The results showed that: (1) The coupling effects of multiple factors yield significant variations when combined in different scenarios. Managers can optimize ecosystem services by strategically regulating factors within specific areas through the acquisition of various probabilistic distributions and combinations of key factors in positive coupling effect scenarios. The outcome is a positive coupling effect. (2) Among the four main ecosystem services in the Pu'er region, food availability and biodiversity were affected by key transboundary and local factors. This suggests that the coupling of transboundary and local factors is more likely to have a significant impact on these two ecosystem services. Of the 45 combination scenarios on food availability, the majority exhibit a negative coupling effect. In contrast, among the 45 combination scenarios on biodiversity, most scenarios have a positive coupling effect. This indicates that food availability is at a higher risk of being influenced by the coupling effects of multiple factors, while biodiversity faces a lower risk. (3) Transboundary pests & diseases, application of pesticides, fertilizer & filming , population density, and land use were the key factors affecting food availability. Bio-invasion, the normalized differential vegetation index, precipitation, and the landscape contagion index were the key factors affecting biodiversity. In this case, focusing on preventing transboundary factors such as transboundary pests & disease and bio-invasion should be the goal.

(4) Attention should also be paid to the conditions under which these transboundary factors combine with local factors. In the areas where these negative coupling effects occur, enhanced monitoring of both transboundary and local factors is essential to prevent adverse effects.

# INTRODUCTION

The coupling effect (CE) is a widespread phenomenon in natural systems (*Bae et al., 2019*; *Nguyen et al., 2020*). The concept refers to the mutual interactions between objects or multiple bodies, which can lead to increasing or decreasing effects (*Guo et al., 2020a*; *Guo et al., 2020b*; *Hestand & Spano, 2018*). In economic or geographical analysis, CEs are characterized by the interaction of two or more factors that influence the performance of the affected system (*Song et al., 2017*). Numerous studies have concentrated on the effects of coupling or synergistic relationships between these factors or systems (*Ariken et al., 2020*; *Fang, Liu & Li, 2016*). However, the impact of multiple factors on the system under different scenarios is unknown. This creates substantial challenges for the systematic management of the affected system. This phenomenon is also commonly observed when multiple factors affect ecosystem services (ESs). An example of this is when food availability significantly decreases in areas where both water pollution and surface pesticide contamination coexist (*Díaz et al., 2019*). To effectively diminish uncertainty influence, it is crucial to precisely quantify and spatialize the impact of these types of multiple factors on ecosystem services.

Factors selected for analysis of CEs on ESs are often local, such as the terrain (*Ganasri & Ramesh, 2016*), soil (*Velasquez & Lavelle, 2019*), biology (*Hu et al., 2012*), climate (*Sintayehu, 2018*), land use (*Hasan et al., 2020*; *Montoya-Tangarife et al., 2017*), or social and economic factors (*Turşie & Perrin, 2020*). However, we must consider transboundary factors such as international trade, poor control of immigration across borders, mass transport, inappropriate land management, and bio-invasion (*López-Hoffman et al., 2010*; *Rodríguez-Labajos, Binimelis & Monterroso, 2009*; *Simberloff et al., 2013*) in border areas. The impacts of these factors on ESs vary in different combination scenarios. To determine these variations, it is essential to construct a network representing the relationship between ESs and the factors affecting them under different probability distributions. This will demonstrate the complex mechanisms of multiple factors affecting ESs.

Some stochastic mathematical techniques have been developed to construct a network for analyzing multiple factors under different probability distributions and how they affect ESs. The Monte Carlo simulation based on the interval chance-constrained programming (MC-ICCP) model has been used to study regional ecosystems under uncertain conditions (*Rosentreter et al., 2021*). However, this method cannot characterize spatial heterogeneity and show the interactions between multiple factors when paired together (*Benke et al.,*

*2018*). Another method, back-propagation artificial neural network (BPANN), can characterize the interactive relationship of said factors by establishing the network structure between these elements and is usually a better fit. However, it may fail to explain the exact relationship between independent and dependent variables (*Benmessahel, Xie & Chellal, 2018*).

A Bayesian network (BN) combined with a geographic information system (GIS) is able to quantify and spatialize the CEs of ESs when using multiple factors in different scenarios. BN is a semi-quantitative statistical model in which the qualitative part of the process analyzes the multifaceted relationship between selected factors and the resulting consequences on the affected systems. Quantitative part refers to the quantification of the presentation of the probability distribution status of each node in the network (*Chen & Pollino, 2012*). By establishing a BN, we are able to adjust the probability distribution of nodes and observe changes in the probability and expected values of ESs in response to these adjustments. This is especially true when multiple factors affect ESs. The variation seen in ESs when these factors coupling differs from the sum of their individual effects. It is critical to calculate the exact values in order to determine the magnitude of the CEs. This provides precise direction for decision makers when creating specific management plans (*Guo et al., 2020a*; *Guo et al., 2020b*). Moreover, the spatial heterogeneity of the CEs in different scenarios can be visualized with the BN-GIS model, which is helpful in the management of ESs.

The CEs of key transboundary and local factors on ESs in border areas will lead to great differences and uncertainties of ESs in the region with the change of the probability of occurrence of each factor. Based on the BN-GIS model, this study constructs the BN of various factors and ESs, sets the arrangement and combination scenarios of key factors under different occurrence probabilities, and quantifies the changes of ESs under different scenarios, so as to reveal and quantify the above CEs. The border area examined in this study is the Pu'er region, which typically shares its borders with three countries in southwestern China.

## MATERIALS & METHODS

### Overview of the study area

#### *Location of the Pu'er region*

The Pu'er region (99°09′–102°19′E, 22°02′–24°50′N) is in Yunnan Province, China. It covers an area of 45,000 km$^2$ and includes nine counties and one district. It is located in the southwestern part of Yunnan Province in Southwest China (Fig. 1) and shares the same border with Myanmar, Laos, and Vietnam. Its sub counties, Lancang, Ximeng, and Menglian have a border with Myanmar with a length of 486 km. Jiangcheng County borders Laos and Vietnam. The China-Laos border and the China-Vietnam border within Jiangcheng County are 116 km and 67 km in length, respectively. The Pu'er region has two national ports and one provincial port: Simao Port, Mengkang Port, and Menglian Port. In 2013, the total cargo volume of these three ports was 374,000 tons. In 2019 this reached 1,049,000 tons, marking a significant increase in the total cargo volume (*China Association of Port-of-Entry, 2014*; *China Association of Port-of-Entry, 2020*).
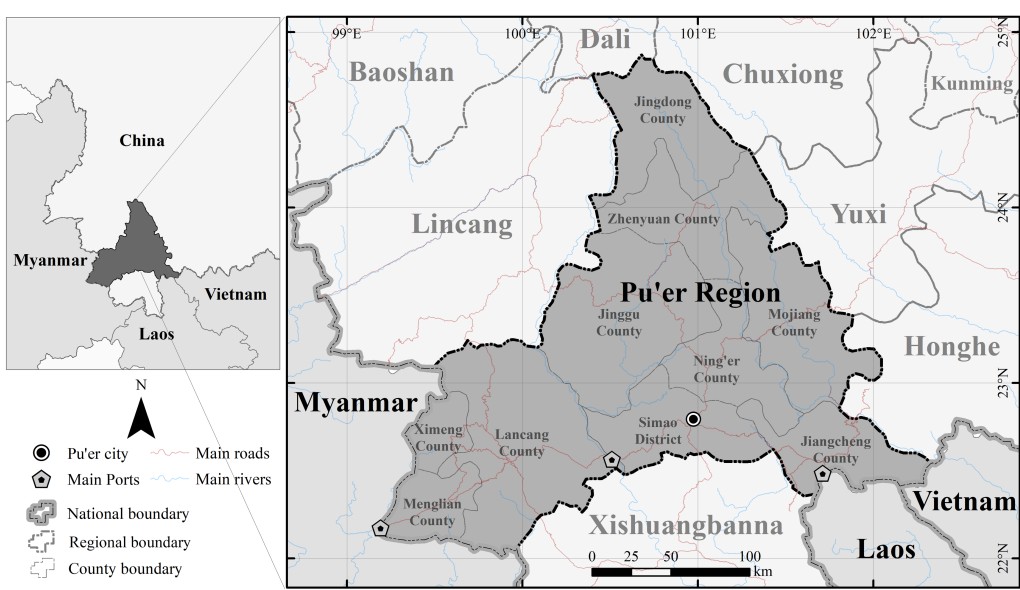

**Figure 1** Map showing the location of the Pu'er region.

### The estimated value of ESs in the Pu'er region administration

The Pu'er region has a subtropical climate influenced by monsoons. The warm, humid air from the Indian Ocean brings a significant amount of precipitation because of the southwest monsoon. Due to ample sunlight and humidity, the forest coverage rate in the Pu'er region is 74.59% (*Ma et al., 2020*). In 2015, the total value of the ESs in the district was 742.87 billion yuan; the value per unit area was estimated to be 0.163.7 billion yuan/km2 and the per capita value was 280.52 thousand yuan. *Ren et al. (2021)* reported that the value of the Pu'er region ranked first in Yunnan Province in terms of the total ES value. Provisioning services account for 3.71% of the total value at 27.53 billion yuan/year. Regulating services account for 95.13% at a value of 706.82 billion yuan/year. Cultural services account for 1.16% at a value of 8.64 billion yuan/year.

### Transboundary factors and their CEs with local factors

By the end of 2018, the fall armyworm (*Spodoptera frugiperda*), originally found in Myanmar, had infested the Pu'er region on a large scale. This insect caused significant damage to farmland, resulting in reduced yields of major crops such as maize (*Wu, Jiang & Wu, 2019*). This reduction was further affected by a strengthened southwest monsoon in southwest Yunnan. The fall armyworm (*S. frugiperda*), yellow-spined bamboo locust (*Ceracris kiangsu Tsai*), and red imported fire ant (*Solenopsis invicta Buren*), which possess strong dispersal abilities, have harmed agriculture and human welfare in many counties in the Pu'er region through their direct impact as transboundary pests and carriers of disease (*Chen, Zhang & Chen, 2022*; *Xiao, Zhou & Quan, 2009*; *Zhang et al., 2019*).

In addition, as cross-border trade increases between Yunnan Province and the neighboring countries (*ASEAN Secretariat, 2020*), plants, animals, and microorganisms have been brought into the Pu'er region during trade and cultural exchanges. This has

posed a greater threat of bio-invasion to the local ecosystem (*Zong, Du & Huang, 2020*). Road construction development has exacerbated the situation (*Xie et al., 2021*). This is an example of the CEs' ability to impact biodiversity (*Yin et al., 2020*).

## Research idea

In this study, we define the term CE as a synthesis of transboundary and local factors that can either increasing or decreasing their total impact on ESs. As a result, CEs have different influence when compared to the effects caused by a single factor alone.

The hypothesis of the study is that the CE of multiple factors under different occurrence probabilities and different combination scenarios will cause great changes in the corresponding ES. Then, we can quantify this coupling effect by observing the values changes of the ES under different occurrence probabilities and different permutation combinations of its key factors.

In addition, when there is a difference between the amount of change generated by the multi-factor CE on the corresponding ES and the sum of the amount of change generated by the single factor on the corresponding ES, we can use this difference to estimate whether the CE will have a positive or negative effect on the corresponding ES.

To investigate the CEs of the key factors on ESs in the Pu'er region, we divided the analysis into three steps, as illustrated in Fig. 2.

In the first step, according to the ecological and environmental characteristics of the Pu'er region (*Pu'er Municipal Environmental Protection Bureau, 2020*), we identified four main types of ESs: food availability (provisioning services), tourism & recreation (cultural services), soil conservation (regulating services), and biodiversity (supporting services). We identified the main factors (transboundary and local) of each type of ES based on a literature review and by consulting local experts. The logical relation network applicable to the BN was simulated using these factors.

In the second step, each node (including each factor and ES) was evaluated and gridded as described in both the 'Valuation of ESs and factors' section and obtained the conditional probability table (CPT) according to BN baseline training (*Feng et al., 2021*). We used the sensitivity analysis of the BN to determine the key factors and their influencing pathway on each corresponding role in the ES.

In the third step, we estimated the CEs of transboundary and local key factors on ESs. Initially, we estimated the expected value (which is a better fit than probability) for each ES under each key factor by adjusting its probability distribution in the BN. In this way, we obtained the individual influence of each key factor on the corresponding ES. Next, we permuted the transboundary and local key factors in different combinations and simultaneously adjusted the probability distribution of these factors in different combination scenarios. This allowed us to determine the expected value variations (EVVs) of the ESs under multiple factors coupling. Within the same combination scenario, we calculated the total effect by summing the EVVs of the ESs under the influence of each key factor. Ultimately, the CEs were calculated as the difference between the of the ESs under multiple factors coupling and the total effect. Based on the changing circumstances in which ESs are affected by CEs, it is essential to advocate for positive CEs and pay special

[1] The basic geographic information associated with the ESs, such as natural and cultural tourism, land-use type, topography, protected area, and others are shown in Appendix A, Fig. A1.

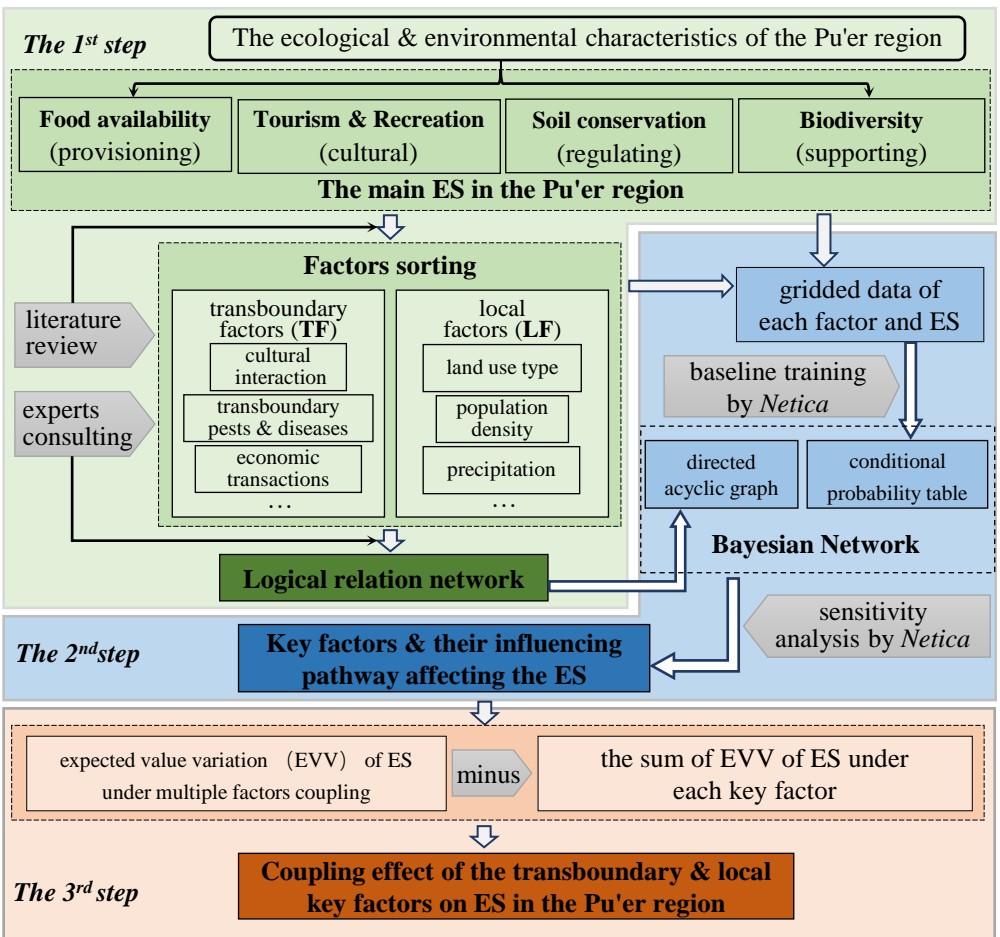

**Figure 2** **The technical process of the study.**

attention to negative CE impacts, especially when formulating corresponding management measures. From a spatial perspective, regions with adverse CEs should be the primary focus for intervention strategies.

## Logical relation networks
### Selection of the main types of ESs in the Pu'er region

Based on the characteristics[1] of the Pu'er region (*Pu'er Municipal Environmental Protection Bureau, 2020*), four main types of ESs (*Costanza et al., 1997*; *Xie et al., 2015*) were selected to represent all of the ESs in the study area. The following items were considered:

- Food availability (FA)

Farmland accounts for 12.4% of the total area (*Pu'er Municipal Environmental Protection Bureau, 2020*); it is the main source for food production. Food availability is the main category within the provisioning services.

- Tourism & recreation (TR)

The Pu'er region has plentiful tourism resources. According to the Institute of Geographic Sciences and Natural Resources, Chinese Academy of Sciences (https://www.resdc.cn/), the Pu'er region contains 397 natural and cultural tourist attractions. Tourism & recreation represent the main source of cultural services.

- Soil conservation (SC)

This region has a complex geographical topography with high mountains and deep valleys. The Ailao Mountain, Wuliang Mountain, Weiyuan River, Xiaohei River, and Amo River experience different degrees of deterioration and soil and water loss. The strategic environmental assessment of the Pu'er region also showed that soil conservation is the main tool used to assess ecological areas. Soil conservation represents the main regulating service in the study area.

- Biodiversity (BD)

Forest covers 74.59% of the Pu'er region. The region contains two national, five provincial, and seven county level nature reserves. It represents one of the most abundant and biodiverse regions in China (*Myers et al., 2000*). Biodiversity is the main supporting service in the study area.

### Factor sorting

After determining the main ESs in the Pu'er region, we chose 15 factors, as shown in Table 1 (all abbreviations for ESs and factors mentioned later are included). Transboundary factors are marked as T and local factors are marked as L (see Appendix A: Table A1 for a detailed description of these factors). Transboundary factors refer to the factors that cross geographical or political boundaries and originate from ecological, economical, societal, and cultural elements. These have direct and indirect impacts on ecosystem services (*Liu et al., 2020*; *Mason et al., 2020*). As can be seen from Table 1, cultural exchanges, economic and trade exchanges, land use, population, and other factors impact the various types of ESs, indicating that there is a complex logical network structure between these factors and the ESs.

It should be noted that the transboundary pests and diseases factor was distinguished from the species invasion factor. This was due to the fact that transboundary pests and diseases had a greater impact on farmland ecosystems, while other types of species invasion (such as invasive species like Mexican sunflower (*Tithonia diversifolia*) which is spread along roads and during cultural exchanges) mainly affect biodiversity.

### Establishment of the logical network from correlations of the factors and the ESs

To characterize the logical network between the factors and the ESs with more accuracy, we invited five experts for the criteria and carried out three rounds of consultations with these experts. We identified the factors of each ES using the Delphi method (*Vogel et al., 2019*), and combined logical networks built by experts with the existing literatures (*Dang et al., 2019*; *Feng et al., 2021*) to establish the influencing pathways of each factor on the corresponding ES which reveals the relationship of network.

**Table 1   Transboundary and local factors affected the main ESs in the Pu'er region.**

| Main ESs | Factors (Abbreviation, Transboundary/Local Factor) |
|---|---|
| Food availability (FA) | economic transactions (ET, T); cultural interaction (CI, T); transboundary pests and diseases (PD, T); land use (LU, L); population density (POP, L); application of pesticide, fertilizer, and filming (AP, L); standardized precipitation index (SPI, L); transportation accessibility (AC, L); slope (SLO, L); precipitation (PRE, L); temperature (TEM, L). |
| Tourism & recreation (TR) | economic transactions (ET, T); cultural interaction (CI, T); transportation accessibility (AC, L); slope (SLO, L); population density (POP, L); land use (LU, L); protected area coverage (PRO, L). |
| Soil conservation (SC) | economic transactions (ET, T); cultural interaction (CI, T); slope (SLO, L); transportation accessibility (AC, L); population density (POP, L); temperature (TEM, L); precipitation (PRE, L); land use (LU, L); standardized precipitation index (SPI, L); protected area coverage (PRO, L). |
| Biodiversity (BD) | economic transactions (ET, T); cultural interaction (CI, T); transboundary pests & diseases (PD, T); bio-invasion (INV, T); land use (LU, L); population density (POP, L); transportation accessibility (AC, L); slope (SLO, L); precipitation (PRE, L); temperature (TEM, L); protected area coverage (PRO, L); normalized differential vegetation index (NDVI, L); landscape contagion index (CON, L). |

## Key factors and establishment of their influencing pathways
### Valuation of ESs and factors

The data used in this study were selected (see Appendix A, Table A1), and we have selected 2020 as a reference point in our analysis. Some data of the ESs and factors, such as BD, SC, and INV were evaluated based on cumulative data over multiple years or were simulated from public datasets (see Appendix B).

To facilitate training at each node in the BN, we gridded and divided the territory of the Pu'er region into over 1,000 square units of equal size.

### Building BN-GIS model for the quantification of the ESs and factors

BN is a graphical probabilistic model with two important components. The first component is the directed acyclic graph (DAG), which links the child nodes to the parent nodes using a set of arrows representing the causal relationships between connected nodes (*Landuyt, Broekx & Goethals, 2016*). The DAG is the part of BN used for qualitative analysis, which can reveal the logical relation networks of the study. The second component is the conditional probability tables (CPTs) of each node, which quantify the relationship among the variables and specify the state of the parent node in order to determine the degree of belief for a particular state (*Forio et al., 2020*; *Pollino et al., 2007*). The CPTs are the part of BN used for quantitative analysis, which is calculated by the baseline training. It supports the sensitivity analysis and the probability distribution adjustment of each node in the network (*van der Gaag, 1996*).

In this study, we established the BN-GIS model using Netica software, which is a specific program used widely in the construction and simulation of BN (*Johnson, Low-Choy & Mengersen, 2012*; *Smith et al., 2007*). We used GIS mainly for assisting in the acquisition of baseline training sample data and for the spatial display of the simulation results (*Guo et al., 2020a*; *Guo et al., 2020b*; *Jensen & Nielsen, 2007*).

### Sensitivity analysis for identifying the key factors and their pathway

After baseline training, the conditional probabilities of the nodes were calculated. The sensitivity analysis was used to calculate the reduction in the variance of the real value of the target node, which was based on the conditional probability of each node in relation to other nodes (*Pearl, 1988*).

The main ESs of the Pu'er region were set as target nodes. The main ESs comprise continuous variable nodes. The degree to which other nodes influenced the target nodes was estimated according to the magnitude of the variance reduction (*Wu et al., 2021*). This was then used to select the key factors. We identified the factor nodes by withdrawing the ES nodes. The factor nodes with the top 25% variance reduction values were determined to be the key factors using the quartile method (*Langford, 2006*).

The influencing pathways of each key factor were determined by using the key factor nodes as the starting points and the corresponding ESs nodes as the ending points in the logical relation network.

### Application of Netica

In this study, Netica was used to integrate the logical relationship networks of the four ecosystem services into a DAG and to preprocess prior data for the generation of CPTs. These CPTs, along with the DAG, collectively form the Bayesian network for the ESs. Netica was also used for sensitivity analysis of the four ESs to identify the key factors and influencing pathways. Furthermore, it allows for the adjustment of probability distributions of different factors to obtain posterior probabilities for the ESs.

## CEs of the key transboundary factors and local factors
### Methodology for the determination of CEs

The formula for CEs is shown in Eq. (1), which was developed based on the physical chemistry of the multiple factors effect (*Hestand & Spano, 2018*).

$$\Delta ES_{CT} - \sum_{1}^{n} \Delta ES_{T_f} \begin{cases} > 0 & (Positive\ coupling) \\ = 0 & (No\ coupling) \\ < 0 & (Negative\ coupling) \end{cases} \tag{1}$$

In Equation (1): when a group of factors $f_1, f_2 \ldots f_n$ ($n \geq 2$) acts together on an ES, the CEs on the ES is expressed as $CT_{(f_1, f_2 \ldots f_n)}$; the individual action of each factor on this ES is $T_{f_1}, T_{f_2} \ldots T_{f_n}$.

The variation of $CT_{(f_1, f_2 \ldots f_n)}$ is $\Delta ES_{CT}$. The total ES variation, with $T_{f_1}, T_{f_2} \ldots T_{f_n}$, is $\sum_{1}^{n} \Delta ES_{T_f}$. $\Delta ES_{CT}$ and $\sum_{1}^{n} \Delta ES_{T_f}$ are related as shown in Eq. (1).

A positive result indicates that the CEs can increase the total effect of each factor on the corresponding ES, while a negative result indicates that the CEs can decrease the total effect of each factor on the corresponding ES.

*Variations of the expected value of ESs*

Using BN can establish a quantitative correlation between multiple factors and the corresponding ES. When we adjust the parametric probability of a factor in a certain state, that of its corresponding ES in each state will also change. This is considered the ES's response to a single factor. When we simultaneously adjust the parametric probability of multiple factors in a certain state, the parametric change of the ES in each state is seen as a response to the CEs of multiple factors. When this response is applied to each unit in a geographic space, there are EVVs of the ESs.

Therefore, when the high state probability of the factors is adjusted to 100 simultaneously, $\Delta ES_{CT}$ is the mean value of the EVV of the corresponding ES for each grid of the study area under the CE of multiple factors. $\sum_1^n \Delta ES_{T_f}$ is the sum of the mean values of the EVV of the ES for each grid of the study area affected by each factor in the same group above, when one adjusts the highest probability of each factor to 100.

Several permutation and combination scenarios of transboundary and local key factors were used to calculate the CE of multiple factors in different scenarios. Using this approach, we can identify which positive CEs should be promoted and which negative CEs should be closely monitored or addressed through the implementation of appropriate management measures. In this way, managers can develop effective strategies for the various scenarios in which ESs are influenced by CEs.

*The expression of spatial variation of the EVVs with that of ESs in the main scenarios*

Spatial variations of ESs in some scenarios can effectively illustrate the ecological risk and ESs management needs of each region. The EVVs of ESs were calculated in each grid in the Pu'er region using Netica software with multiple factors combination scenarios. These grids were expressed spatially using GIS to present the spatial heterogeneity of the corresponding ES. By depicting the spatial heterogeneity of the EVVs of ESs, we can pinpoint potential areas where negative CEs can occur. This can assist managers in adjusting management strategies for specific locations.

## RESULTS

### Logical relation network of the main ESs

The logical relation network of the main ESs in the Pu'er region was established after consulting with experts at three different times (Fig. 3).

### Key factors and their pathways
*Baseline training*

Once the DAGs of the nodes had been properly established, we integrated four logical relation networks into the initial network structure of BN. After evaluating and gridding each node in the network, the CPT of each node in the network (Fig. 4) was estimated using baseline training.

Figure 4 shows that the nodes of FA, TR, SC, and BD had high probability distributions (4.58%, 31.5%, 34.9%, and 21.7%, respectively), moderate probability distributions
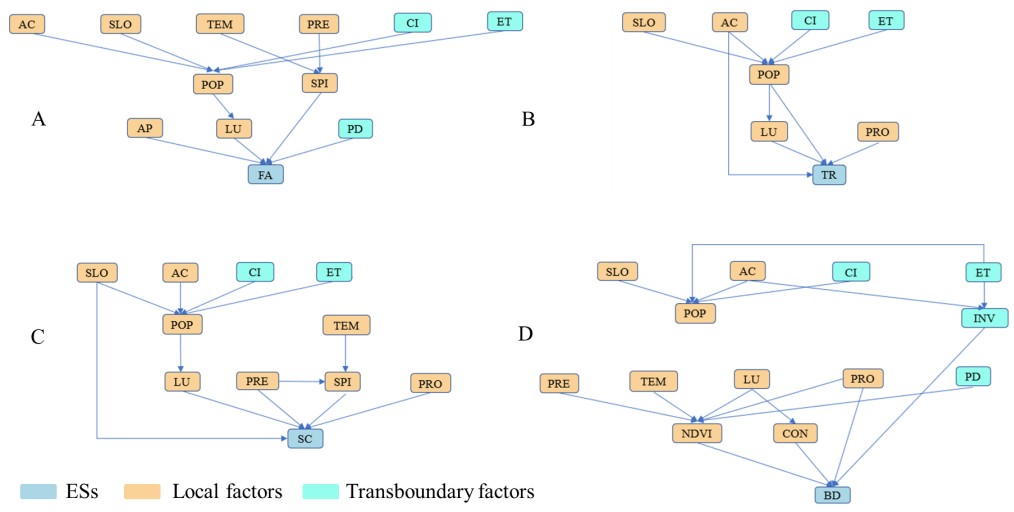

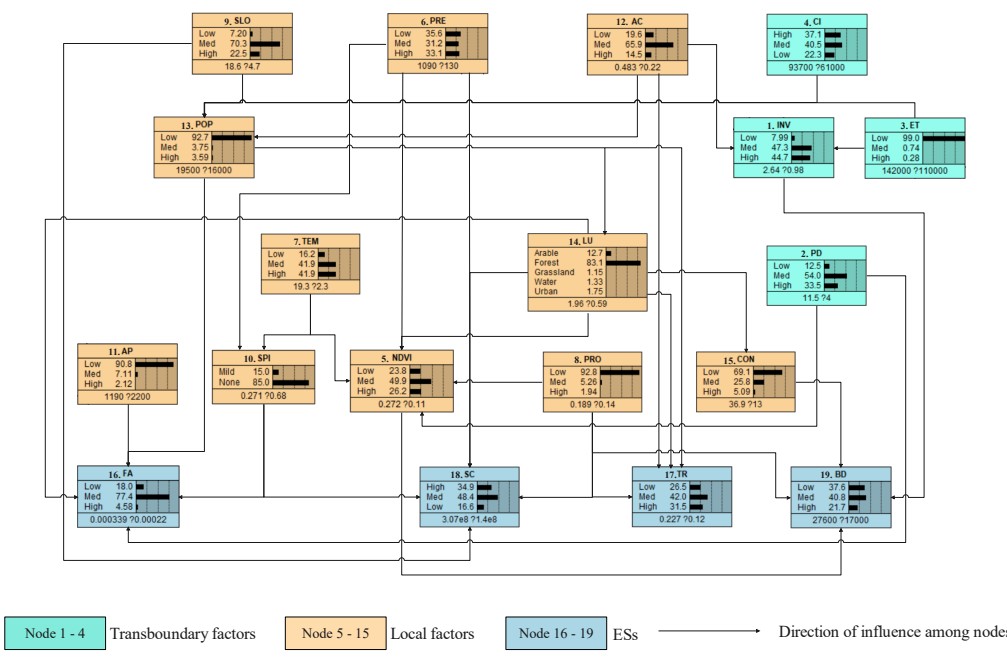

**Figure 3** **Logical relation network s of the main ESs in the Pu'er region.** (Refer to Table 1 for abbreviations.)

**Figure 4** **Current distribution probability for all nodes of the network.** Analysis of the CE from transboundary and local factors acting together on the ESs (endpoints of the network) in the Pu'er region. Each parameter (box) had potential states, *i.e.,* low, medium, high, or states using other classification criteria. The distribution shown in each box is the value of the quantified factors distributed for that parameter, with the expected value ±the standard deviation shown below the distribution. The arrow connecting the two nodes means that the two nodes are causally related, which is fully consistent with the linkages between the nodes in Fig. 3. (Refer to Table 1 for abbreviations.).

**Table 2  Results of BN sensitivity analysis.[a]**

| FA | | TR | | SC | | BD | |
|---|---|---|---|---|---|---|---|
| Node (T/L) | Variance reduction (%) | Node (T/L) | Variance reduction (%) | Node (T/L) | Variance reduction (%) | Node (T/L) | Variance reduction (%) |
| FA | 100 | TR | 100 | SC | 100 | BD | 100 |
| **AP**(L) | 1.7300 | **AC**(L) | 20.8000 | **PRE**(L) | 7.8600 | **INV**(T) | 1.5000 |
| **POP**(L) | 1.3400 | **LU**(L) | 0.9100 | **SLO**(L) | 1.1600 | **NDVI**(L) | 0.9640 |
| **LU**(L) | 0.8730 | **PRO**(L) | 0.3720 | **LU**(L) | 1.0600 | **PRO**(L) | 0.0710 |
| **PD**(T) | 0.5380 | **POP**(L) | 0.0735 | **PRO**(L) | 0.6410 | **CON**(L) | 0.0527 |
| AC(L) | 0.0721 | INV(T) | 0.0730 | NDVI(L) | 0.1000 | PD(T) | 0.0278 |
| ET(T) | 0.0693 | CI(L) | 0.0546 | POP(L) | 0.0782 | PRE(L) | 0.0130 |
| SLO(L) | 0.0602 | SC | 0.0426 | TR | 0.0234 | TEM(L) | 0.0129 |
| SPI(L) | 0.0573 | FA | 0.0391 | CON(L) | 0.0209 | AC(L) | 0.0080 |
| CON(L) | 0.0303 | CON(L) | 0.0263 | SPI(L) | 0.0107 | LU(L) | 0.0071 |
| SC | 0.0071 | SLO(L) | 0.0181 | FA | 0.0077 | POP(L) | 0.0038 |
| CI(T) | 0.0069 | NDVI(L) | 0.0123 | AC(L) | 0.0057 | ET(T) | 0.0007 |
| NDVI(L) | 0.0049 | ET(T) | 0.0041 | ET(T) | 0.0030 | FA | 0.0003 |
| TR | 0.0045 | BD | 0.0030 | BD | 0.0009 | SC | 0.0002 |
| PRE(L) | 0.0015 | AP(L) | 0.0000 | CI(T) | 0.0005 | SPI(L) | 0.0002 |
| TEM(L) | 0.0004 | SPI(L) | 0.0000 | AP(L) | 0.0000 | SLO(L) | 0.0001 |
| BD | 0.0002 | PD(T) | 0.0000 | INV(T) | 0.0000 | TR | 0.0001 |
| INV(T) | 0.0001 | TEM(L) | 0.0000 | PD(T) | 0.0000 | CI(T) | 0.0000 |
| PRO(L) | 0.0000 | PRE(L) | 0.0000 | TEM(L) | 0.0000 | AP(L) | 0.0000 |

**Notes.**

[a]The quartile method (*Langford, 2006*) was used to screen the key factors. In this study, the non-ES nodes with the greatest change reduction in the top 25% were selected as the key factors affecting the corresponding ES. These factors are shown in **bold** in the table.

(77.4%, 42.0%, 48.4%, and 40.8%, respectively), and low probability distributions (18.0%, 26.5%, 16.6%, and 37.6%, respectively).

### Key factors and the main influencing pathways

After the sensitivity analysis of each main ES, the variance reduction of each node to the corresponding ES was calculated (Table 2). After eliminating the ES node of FA, TR, SC, and BD, we used the quartile method (*Langford, 2006*) to identify the key factors of each ES. There are four key factors for each ES (see abbreviations in bold in Table 2). The influencing pathway of the key factors was then determined (Fig. 5). There are three key influencing pathways of FA and four main influencing pathways of TR, SC, and BD.

As shown in Fig. 5, the key factors for FA and BD include both local and transboundary factors, while SC and TR consist solely of local key factors.

## CEs of the transboundary and local key factors

We chose FA and BD with their key factors as targets to analyze the CEs, as they contained both transboundary and local key factors.
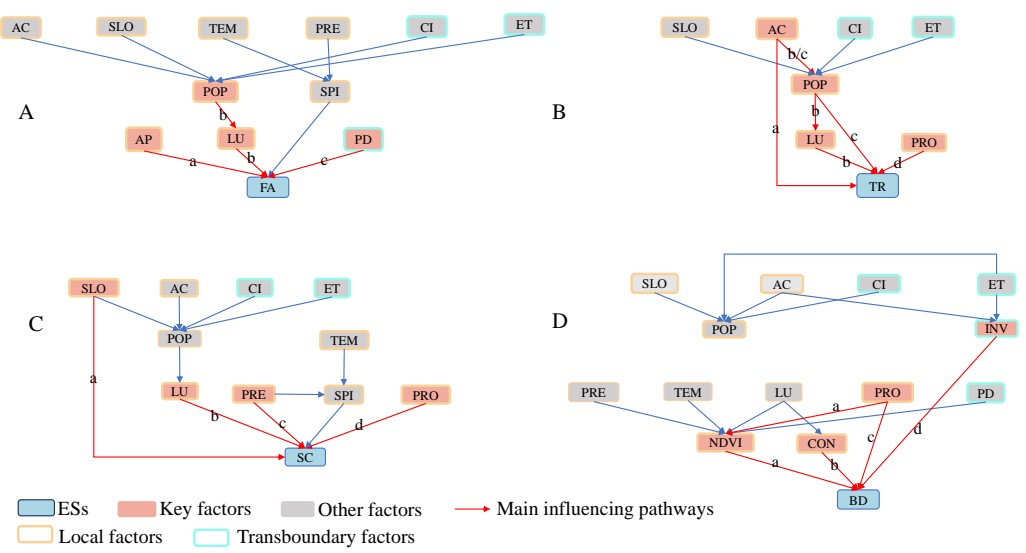

**Figure 5** **Key factors and main influencing pathways.** The pathway with the same lowercase letter in each graph is a main influencing pathway affecting the corresponding ES. For example, in (A) POP → LU → FA is the influencing pathway of FA marked as the letter "b". (Refer to Table 1 for abbreviations.)

### CEs on FA

The key transboundary factor of FA was PD. The key local factors were AP, POP, and LU. Because POP and LU were on the same key pathway (pathway "b" in Fig. 5A) and POP was the parent node of LU, this study selected POP in FA as the main node in pathway "b" to calculate the CEs of the factors.

The transboundary factor PD was combined with the local factors AP and POP. The CE values of the factors on FA were calculated under each combination scenario. As shown in Fig. 6, the number of scenarios with a positive CE is significantly less than those with a negative CE. The highest positive CE is about twice as small as the negative CE. This indicates that FA is more likely to be affected by a large negative CE.

### CEs on BD

The key transboundary factor of BD was INV. The key local factors were CON, PRO, and NDVI. Because PRO and NDVI were on the same key pathway (pathway "a" in Fig. 5D) and PRO was the parent node of NDVI, this paper selected PRO as the main node in pathway "a" of BD to be used in the calculation of the CEs of the factors.

The transboundary factor INV was arranged and combined with the local factors CON and PRO. The CE values of the factors on BD were calculated for each combination scenario. As shown in Fig. 7, the number of scenarios with a positive CE is higher than the number of scenarios with a negative CE. The highest positive CE was greater than the number of negative CEs. This indicated that BD was more likely to be affected by a large positive CE.
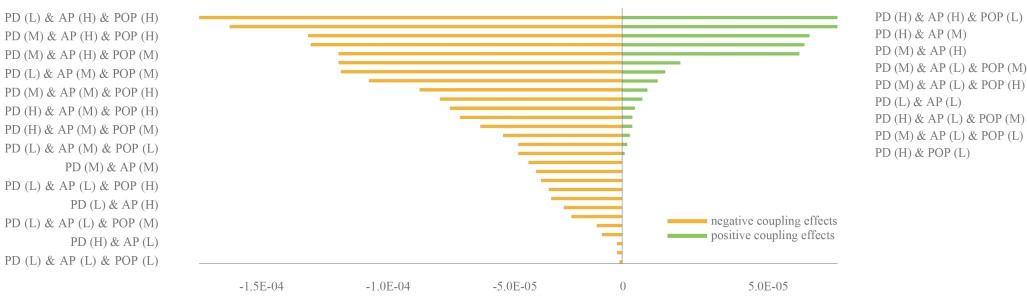

**Figure 6  CEs on FA in different combination scenarios.** The letters H, M, and L in the brackets represent the high/medium/low states of the factors' probability distribution, which are shown in the factors' boxes in Fig. 4. When the probability distribution state was (H), it meant that the probability of this factor in the H state was 100%. The expected value of FA after baseline training was 3.44e−4. (Refer to Table 1 for abbreviations).

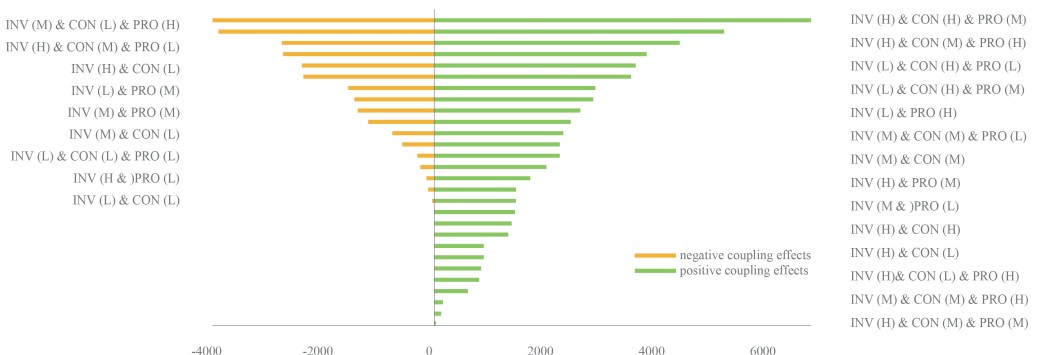

**Figure 7  CEs on BD in different combination scenarios.** The letters H, M, and L in brackets had the same meaning expressed in Fig. 6. The expected value of BD after baseline training was 27,576.8. (Refer to Table 1 for abbreviations).

### Spatial expression of the EVVs of the ESs in the main scenarios

In the transboundary and local factor combination scenarios for FA and BD, only the scenarios with the largest and smallest CEs were screened out for spatial expression, as shown in Fig. 8. PD (H) & AP (H) & POP (L) and PD (L) & AP (H) & POP (H) are the maximum and minimum CE scenarios for FA, respectively. INV (H) & CON (H) & PRO (M) and INV (M) & CON (L) & PRO (H) are the maximum and minimum CEs for BD, respectively.

As seen in Fig. 8, the expected value of FA and BD in each grid shows strong spatial heterogeneity in the Pu'er region. The CE on the ES is reflected in the expected values for all grids in the Pu'er region. The heterogeneity patterns are helpful for all levels of ecological management and can be used to adjust conservation strategies for different geographical features within the region.

For example, in the heterogeneity patterns in Fig. 8B, the strongest negative CE on FA occurs mainly in Lancang County and Mojiang County; this situation should be avoided and prevented in these counties. The heterogeneity patterns in Fig. 8C, the region where

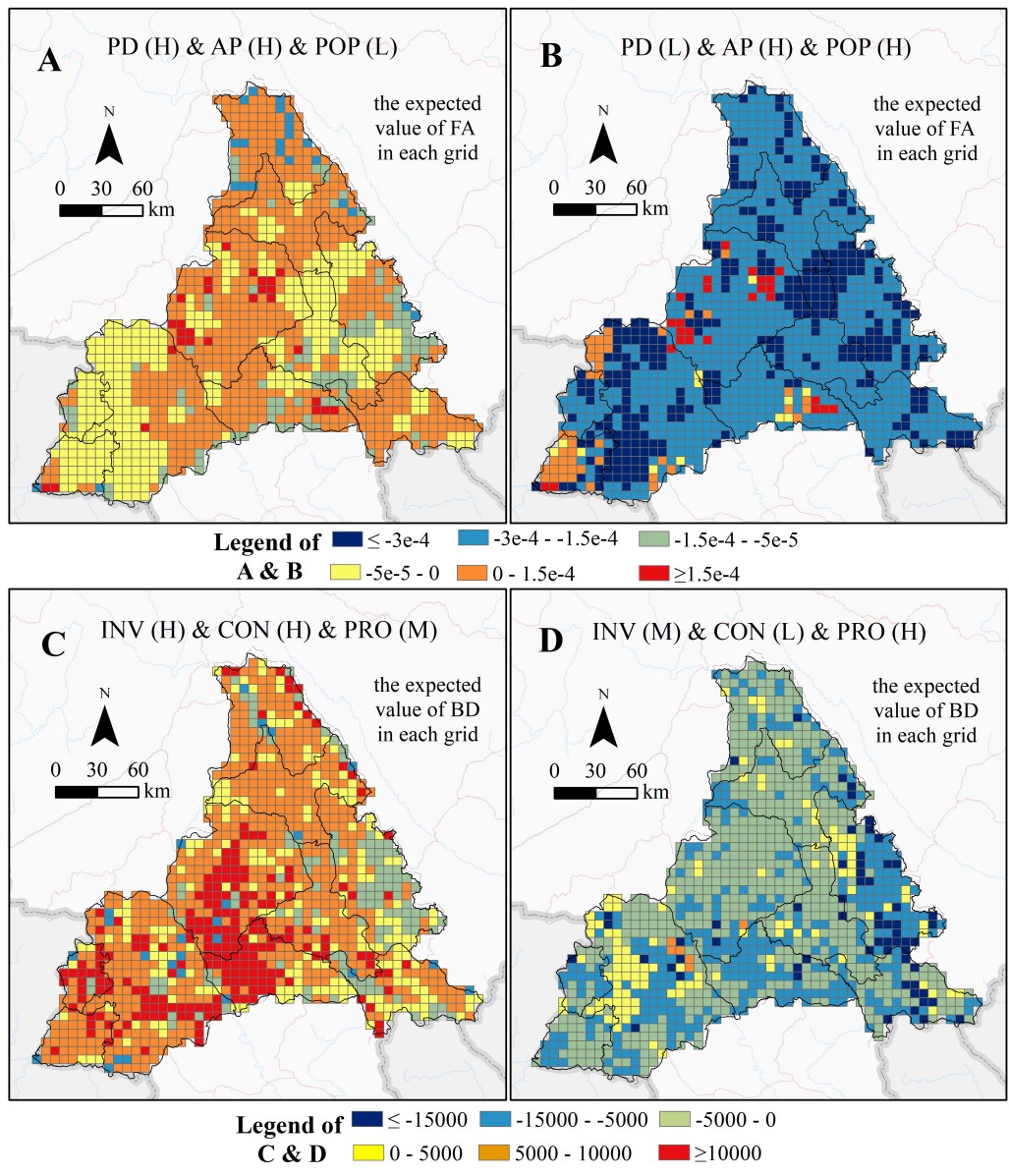

**Figure 8** **Expected values of FA and BD in different scenarios in each grid.** (A) and (B) were the spatial expressions of the max and min CE scenarios on FA, respectively. (C) and (D) were the spatial expressions of the max and min CE scenarios on BD, respectively. The colors from dark blue to red in each grid represent the expected value of FA (A & B) or BD (C & D) from min to max. (Refer to Table 1 for abbreviations.).

BD shows the strongest positive CE, appear mainly in Jingdong County, Simao District, and Lancang County. This is information these counties should take account into when considering conservation strategies.

## DISCUSSION

### Using combination scenarios as precise support for ES management

The prevailing definitions of the CE in previous ES studies focused on the interactions between objects or multiple entities (*Hestand & Spano, 2018*) and the interaction of multiple factors that influence the performance of the affected system (*Song et al., 2017*). In contrast, our study refines the definition of the CE as the effect stemming from the simultaneous effects of multiple factors on ESs. The study revealed the specific scenarios in which ESs exhibit positive or negative CE and highlighted the spatial heterogeneity of these CEs (*Benke et al., 2018*; *Rosentreter et al., 2021*). It is easy for users to apply the trained BN-GIS model to other border areas. The presence of CEs in the Pu'er region has proven that these effects vary when the probability of the factors occurring changes. By calculating the positive and negative CEs in different scenarios, we can devise pertinent management strategies that promote positive CEs. We can also give particular attention to regions exhibiting negative CEs.

Of the spatial heterogeneity observed among CEs on ESs when using 90 multiple factors combination scenarios, FA was more likely to produce negative CEs when subjected to the coupling of three key factors. This negative effect was higher than the positive CE value (Fig. 6), which poses a challenge to the farmland ecosystem and to agricultural development in the Pu'er region. When PD and AP were high and POP was low, the three key factors produced the highest positive CE on FA. This indicates that increasing the application of pesticide, fertilizer, and filming could suppress the effect of transboundary pests & diseases in areas with low population density. In addition, the results also showed that BD was more likely to produce a positive CE when the three key factors were combined. Through the combination scenarios of key factors that generate positive CEs, as seen in Fig. 7, local managers could focus on improving inter-regional connectivity and increasing the size of protected areas as appropriate. Figure 8 shows the spatial distribution characteristics of the CEs of the key factors acting on FA and BD under the two extreme scenarios (scenarios where positive and negative CEs are strongest). It illustrates that there were large spatial differences in the CEs of key factors on the ESs in different scenarios. Managers could develop control measures for these key factors based on such regional differences.

### Transboundary factors vary regionally

This study places more emphasis on transboundary factors, rather than local factors. In total, four transboundary factors were listed: INV, PD, ET, and CI (Figs. 3 and 5). ET, CI, and PD are all parent nodes in the BN, while INV functions as both a child and parent node. This distinction indicates that ET, CI, and PD are a source of transboundary risk, whereas INV represents the interference pressure exerted on the ecosystem by these risks (*Gong et al., 2021*). The sensitivity analysis conducted on the BN model revealed that ET and CI were not key factors and therefore, their influence is limited. On the other hand, PD as a source of risk for transboundary pests and diseases, effectively represents a refined form of bio-invasion. This primarily affects agricultural production and subsequently affects ESs such as FA. According to the results of this study, the risk of bio-invasion is still the main

transboundary factor significantly impacting the local ecosystem. These results align with existing research in this area (*Facon et al., 2006*; *Gilioli et al., 2017*; *Yin et al., 2020*).

The sensitivity analysis showed that ET and CI were not key factors of the four ESs in the Pu'er region. However, it is worth noting that the Pu'er region may not be a typical border area given the extensive transboundary social and ecological impacts associated with the economic and cultural interactions that occur in the region. Therefore, it may not be an ideal region to use for analyzing the influences of ET and CI. *Ala-Hulkko et al. (2019)* investigated the impact national borders have on ESs by restricting the food trade within nation-state borders. Cultural interactions could promote tourism in border areas and contribute to the conservation of BD (*Dunets, Ivanova & Poltarykhin, 2019*). Sometimes CI did not directly affect the ESs. For example, Kiswahili is used as an intercultural communication tool for Kenya-Uganda transboundary trade (*Odhiambo, Losenje & Indede, 2022*). With the aid of CI, ET can indirectly impact the production and livelihood of individuals in border areas, subsequently influencing local ESs (*Ala-Hulkko et al., 2019*). Besides cultural differences, economic or other factors could lead to differences in environmental awareness between the two types of residents living on both sides of the border but in the same natural ecosystem. This could result in estimating the wrong value to be placed on some ecological environments (*Sagie et al., 2013*).

The ET and CI in the Pu'er region had less impact on the ESs mainly because: (1) The total border trade volume was low; for example, the trade volume of the Simao port in the past three years was minimal. The neighboring countries are less developed and China exports to these countries. This leads to a relatively weak economic and trade impact on China. (2) CI of neighboring countries in the Pu'er region was more impacted by the people from neighboring countries engaging in agricultural labor in the Pu'er region. Therefore, it had little influence on the tourism & recreation services in the Pu'er region. (3) The influence of ET and CI often extend from the more developed side to the less developed side of the border, such as when the ET and CI of the United States affects Mexico (*Norman et al., 2012*), Western Europe affects Eastern Europe (*Ala-Hulkko et al., 2019*), and China affects Southeast Asia and North Korea (*Liu et al., 2020*).

In addition, the influence of transboundary factors on ESs decreases with distance, as seen in the PD. (*Wu, Jiang & Wu, 2019*). However, not all transboundary factors are associated with spatial distance, as ET requires more consideration of the economic distance of each location (*Wang et al., 2020*).

## Uncertainty of the study and prospect

Although this study reveals important discoveries, it also has limitations. First, the CEs of FA in the multiple factors combination scenarios have low interpretability because the current data used to quantify factors was under-represented. Second, the BN-GIS model can only provide the expected value of the ESs. This might be different from the true value of the ESs when influenced by the CEs, as the model focused more on the change in the conditional probability of each node. Third, the study focused on the analysis of the CEs between the factors on a regional scale, possibly ignoring transboundary ecological risks affecting the ESs on a smaller scale.

Notwithstanding its limitations, this study is promising. This study did not consider the factors of a single ES (*Dang et al., 2019*; *Guo et al., 2020a*; *Guo et al., 2020b*; *Omer, Pascual & Russell, 2010*) but established a logical network of four types of ESs and their influencing factors. It then analyzed the CEs of multiple key factors on the ESs. The advantage of this approach was that it not only showed the relationship between transboundary and local factors on different ESs but also reflected the complexity of the regional social ecosystem. This makes the simulation more practical. The study also provided a tool for detecting the spatial heterogeneity of the CE on the ES, which could aid local managers in developing management strategies in different regions.

Future research on the trade-offs and synergies between ESs may in fact demonstrate even greater impact. With limited resources, managers must seek to improve the overall ESs rather than a single ES. This means meeting the needs of many stakeholders rather than a single stakeholder. The experimental research results will hopefully serve as useful feedback to guide improvements in ecological risk prevention and management in border areas.

## CONCLUSIONS

The study quantifies the CEs of key factors on ESs in border areas under different scenarios. The results confirm the hypothesis of this study. The impact of transboundary and local key factors on ESs varies greatly under different scenarios. This shows that even if the factors remain the same, only changing the probability of occurrence of each factor can make a huge difference in regional ESs. This poses a great challenge to regional ecological management. Managers should not only identify the factors that affect an ES, but also determine whether the combination of factors under different probability of occurrence has a positive or negative impact on the ES. The positive coupling effect of multiple factors can effectively improve the corresponding ESs. Negative coupling will reduce the corresponding ESs. Based on the method provided in this study, managers can determine the occurrence probability and combination scenario conditions of positive CEs, and guide the management and control of specific influencing factors. Managers can also guide regions with low EVV to improve ESs by controlling the occurrence probability of influencing factors in combination with the spatial pattern characteristics of EVV in positive coupling scenarios.

In addition, the study also analyzed the CEs of transboundary factors and local factors on ESs in border areas. It has a great guiding on the prevention and control of ecological risks in border areas. The key transboundary factors affecting ESs vary across border area. In the Pu'er region, it is noteworthy that PD and INV are pivotal factors among the transboundary influences on ESs. It is imperative to remain vigilant regarding these transboundary ecological risks. The Pu'er region, characterized by frequent trade with neighboring countries, is less impacted by ET and CI. However, ET and CI on ESs should not be considered less significant for other border regions. Typically, the effects of ET and CI tend to radiate from the more developed side to the less developed side of the border.

In this study, while quantifying the multi factor CEs in the border area, there are still some problems such as complex logical network structure and lack of interpretability

of model results. These problems still need to be further optimized and improved by exploring the impact mechanism of ESs. In addition, the trade-offs between ESs in border areas should also be valued and explored. This will be an important part of improving the overall resource utilization efficiency in the region.

## ACKNOWLEDGEMENTS

We thank Dr. Zhiguo Li and Dr. Shaojuan Li at the Yunnan University of Finance and Economics for their assistance in establishing the logical relation network in this work.

### Funding

This work was supported by the Special Project for Basic Research in Yunnan Province (Key Project) (202001BB050073), the Humanities and Social Sciences Youth Foundation, the Ministry of Education of the People's Republic of China (Western and Border Areas Project) (20XJAZH005), the National Natural Science Foundation of China (4226010291), the China-Myanmar Joint Laboratory for Ecological and Environmental Conservation (C176240208), and Scientific Research and Innovation Project of Postgraduate Students in the Academic Degree of YunNan University (KC-23234266). The funders had no role in study design, data collection and analysis, decision to publish, or preparation of the manuscript.

### Grant Disclosures

The following grant information was disclosed by the authors:
The Special Project for Basic Research in Yunnan Province (Key Project): 202001BB050073.
The Humanities and Social Sciences Youth Foundation.
The Ministry of Education of the People's Republic of China (Western and Border Areas Project): 20XJAZH005.
The National Natural Science Foundation of China: 4226010291.
The China-Myanmar Joint Laboratory for Ecological and Environmental Conservation: C176240208.
Scientific Research and Innovation Project of Postgraduate Students in the Academic Degree of YunNan University: KC-23234266.

### Competing Interests

The authors declare there are no competing interests.

### Author Contributions

- Ruijing Qiao conceived and designed the experiments, performed the experiments, analyzed the data, prepared figures and/or tables, authored or reviewed drafts of the article, and approved the final draft.
- Jie Li analyzed the data, prepared figures and/or tables, authored or reviewed drafts of the article, and approved the final draft.

- Xiaofei Liu analyzed the data, prepared figures and/or tables, and approved the final draft.
- Mengjie Li performed the experiments, analyzed the data, prepared figures and/or tables, and approved the final draft.
- Dongmei Lei conceived and designed the experiments, authored or reviewed drafts of the article, and approved the final draft.
- Yungang Li analyzed the data, prepared figures and/or tables, and approved the final draft.
- Kai Wu analyzed the data, prepared figures and/or tables, and approved the final draft.
- Pengbo Du analyzed the data, prepared figures and/or tables, and approved the final draft.
- Kun Ye analyzed the data, authored or reviewed drafts of the article, and approved the final draft.
- Jinming Hu conceived and designed the experiments, authored or reviewed drafts of the article, and approved the final draft.

## Data Availability

The raw measurements are available in the Supplemental File.

## Supplemental Information

Supplemental information for this article can be found online at http://dx.doi.org/10.7717/peerj.17015#supplemental-information.

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
