# Peer review of "Coupling effect of key factors on ecosystem services in border areas: a study of the Pu’er region, Southwestern China"

_PeerJ, doi:10.7717/peerj.17015_

## Round 0.1 · original submission · Major Revisions

The reviewers have raised some significant concerns about the manuscript. There are both methodological and information requirements that need to be clearly outlined. The objectives and hypotheses need to be clearly stated. Some of the data and scripts used in the modeling must be presented clearly to replicate the results obtained. The conclusion is written poorly and needs to provide expanded discussion on the results, especially policy implications. The paper needs a thorough edit for language and writing. There is a clear need to justify why this is unique in contribution compared to what is available in the literature.

**Language Note:** The Academic Editor has identified that the English language must be improved. PeerJ can provide language editing services - please contact us at copyediting@peerj.com for pricing (be sure to provide your manuscript number and title). Alternatively, you should make your own arrangements to improve the language quality and provide details in your response letter. – PeerJ Staff

Reviewer 1 ·

Basic reporting

no comment

Experimental design

no comment

Validity of the findings

no comment

Additional comments

The study area is typical, and the content of the study also has a certain degree of innovation. The review comments and suggestions are as follows.
1. In Section 2.1.1, how did the author obtain the statistical data for the year 2019 from the Statistical Yearbook of 2019? We recommend that the author verify the publication date of the Statistical Yearbook.
2. In Section 2.1.2, the sum of the proportions of supply services, regulating services, and cultural service values does not add up to 100%. Please verify and make the necessary corrections.
3. In this article, the author did not provide a detailed explanation of the concept of transboundary factors, the criteria for selecting transboundary factors, and the necessary rational analysis. Furthermore, the author should provide comprehensive justification for the selection of the four ecosystem services, including an analysis of their typicality, representativeness, and uncertainty in the study area. Additionally, the calculation methods and processes for these four ecosystem services are not clearly explained in the text.
4. Is the method used to calculate the coupling effects of transboundary factors and local factors cited from a reference or developed by the author? If it is cited, the corresponding references need to be properly cited.
5. The discussion section is not comprehensive enough. It is suggested that the author conduct a thorough comparison of this study with current relevant research to highlight the innovation of this study. Furthermore, it is necessary to further explore the shortcomings in this study, as well as the uncertainties in the research results.

Reviewer 2 ·

Basic reporting

no comment

Experimental design

no comment

Validity of the findings

no comment

Additional comments

The novelty of the manuscript lies in quantifying the coupling effect of transboundary and local factors on ecosystem services, which provides a new way for the research of coupling effect. Accordingly, negative and positive coupling effect were calculated in food availability and biodiversity services. The novel findings can benefit the ecosystem protection and management in border areas.

However, there are still some flaws to be amended.
1. The manuscript is poorly readable. There are many abbreviations of terms (4 ecosystem services and 5 factors) in text and figures, the readers have to find them back and forth in manuscript. CCE in line 263 should be given full name (Is it should be CE?). Appendix A: Table A2 mentioned in line 287 and 289 can’t be found in Appendix section.
2. It is recommended to introduce the specific functions of Netica software.
3. The descriptions in line 290 to 291 don’t match with those in Fig. 5.
4. It is recommended to give reasons for choosing FA and BD as target to analyze the CEs rather than all four ecosystem servics.
5. It is unclear in line 365.
6. A new method was proposed to quantify the CEs, so uncertainties and limitations should be added in discussion section to make conclusions convincing.
7. Zero value was suggest to take into account in data grading of Fig. 8 A and B.

---

## Round 0.2 · Minor Revisions

Minor revisions are needed for better clarity in the research. Please address the following:

- List general and specific objectives of your research at the end of the introduction section

- List hypotheses that you are testing - this should follow the specific objectives you list

- The conclusion section needs more information on the weaknesses and caveats of the study

- Summarize policy implications of your results clearly for regional policies and governance.

Reviewer 1 ·

Basic reporting

no comment

Experimental design

no comment

Validity of the findings

no comment

Additional comments

The revised manuscript submitted by the authors was carefully revised in accordance with the expert comments and suggestions, and the quality of the article has been significantly improved. However, the reference format is recommended to be revised according to the requirements of the journal.

Reviewer 2 ·

Basic reporting

no comment

Experimental design

no comment

Validity of the findings

no comment

Additional comments

The authors have amended all the flaws I pointed out. The manuscript has met the standard of publication.

---

## Round 0.3 · accepted · Accept

The authors have addressed the comments made. The manuscript is ready for publication.